# Radiation Protection of a 3D Computer Tomography Scanning Workplace for Logs—A Case Study

**DOI:** 10.3390/s23218937

**Published:** 2023-11-02

**Authors:** Tomáš Gergeľ, Juraj Hamza, Vojtěch Ondrejka, Miroslav Němec, Miroslav Vanek, Jennifer Drugdová

**Affiliations:** 1National Forest Centre, Forest Research Institute, T. G. Masaryka 22, 960 01 Zvolen, Slovakia; tomas.gergel@nlcsk.org (T.G.); juraj.hamza@nlcsk.org (J.H.); jennifer.drugdova@invictum.sk (J.D.); 2Faculty of Forestry and Wood Technology, Mendel University in Brno, Zemědělská 1665/1, 613 00 Brno, Czech Republic; vojtech.ondrejka@mendelu.cz; 3Faculty of Wood Sciences and Technology, Department of Physics, Electrical Engineering and Applied Mechanics, Technical University in Zvolen, T.G Masaryka 24, 960 01 Zvolen, Slovakia; 4Faculty of Ecology and Environmental Sciences, Department of Environmental Engineering, Technical University in Zvolen, T.G Masaryka 24, 960 01 Zvolen, Slovakia; vanek@tuzvo.sk

**Keywords:** CT scanner, safety, leakage radiation, risk, measurement, radiation protection

## Abstract

Despite its undeniable advantages, the operation of a CT scanner also carries risks to human health. The CT scanner is a source of ionizing radiation, which also affects people in its surroundings. The aim of this paper is to quantify the radiation exposure of workers at a 3D CT wood scanning workplace and to determine a monitoring program based on measurements of ionizing radiation levels during the operation of a CT log scanner. The workplace is located in the Biotechnology Park of the National Forestry Centre. The ionizing radiation source is located in a protective cabin as a MICROTEC 3D CT machine with an X-ray lamp as X-ray source. The CT scanner is part of the 3D CT scanning line and its function is continuous quality scanning or detection of internal defects of the examined wood. The measurement of leakage radiation during scanning is performed with a metrologically verified meter. The measured quantity is the ambient dose equivalent rate ${\dot{H}}^{*}\left(10\right)$. The results of the measurements at the selected measurement sites have shown that, after installation of additional safety barriers, the CT scanner for the logs complies with the most strict criteria in terms of radiation protection. Workers present at the workplace during the operation of the CT scanner are not exposed to radiation higher than the background radiation level.

## 1. Introduction

In recent decades, there has been a growing development of non-destructive quality testing of various materials. This scientific field is called defectoscopy. The role of defectoscopy is to detect defects in a material or product using a set of methods or by monitoring physical parameters. Based on the type of waves that interact with the material, these methods are divided into ultrasonic, infrared, visual and X-ray, and magnetic defectoscopy.

X-ray defectoscopy has started to be applied and developed especially in the medical field of radiology, where X-rays are used to scan living structures. By improving this concept and using computer technology, the first commercial medical CT scanner was developed in 1972 [1]. With gradual development, X-ray scanners also began to find applications in industry for checking defects in castings and welds as well as for scanning baggage at airports. The first tests and modifications for non-medical use of a CT scanner began at the Los Alamos Laboratory, where a research and development program was established for this purpose in 1977 [2]. Today, CT scanners have a wide range of applications mainly in research, reverse engineering, measurement and digitization [3]. An example is the use of CT scanning to detect the distribution of rejuvenation products of porous asphalt [4]. The authors Liang et al., 2020, used industrial CT scanning technology to test the compaction quality of cement-stabilized macadam [5]. The use of CT scanning in investigating the health status of tree seedlings is interesting [6]. A case study from 2016 was undertaken to compare a medical CT scanner with an industrial CT scanner [7].

In the 1980s, Microtec began to develop and use computer tomography technology to scan tree trunks. The aim was to know the internal features of the wood before cutting plans were made. Today we can say that the company has been successful in its efforts and nowadays there are several CT scanners for logs in operation and installed in wood processing plants all over the world. This basic idea has enabled the development in the application of CT scanning in the forestry and timber sector. CT scanners can be used to optimize the position of a trunk during cutting. Research has confirmed a 13% increase in economic value [8]. The application of fast and continuous scanning of timber logs was demonstrated by Ursella et al. The scanning speed reached up to 160 m/min [9]. Another example is the demonstration that the introduction of furniture production optimization using CT scanning can increase the economic value by 11% [10]. The introduction of CT scanning technology is also related to the development of software tools for the automatic detection of wood defects, the use of neural networks [11] and the principles of computer vision [12,13].

Despite its undeniable advantages, the operation of a CT scanner also carries risks to human health. A CT scanner is a source of ionizing radiation and working near it is an activity causing radiation exposure. Today, industrial CT scanning technology is so well developed, we can say that it is a very safe technology using which a worker is exposed to a very small dose of radiation [14,15]. It has been shown the benefits obtained outweigh the risks when CT scanners are used in medicine [16,17,18]. The question is whether this is also the case for industrial applications.

Radiation protection is a system of technical and organizational measures to protect people from the effects of ionizing radiation [19]. The system of variables used in radiation protection is dynamically evolving and current information is published in the ICRU (International Commission on Radiation Units and Measurements) and ICRP (International Commission on Radiological Protection) materials. The quantities used can be divided into operational quantities, radiation protection quantities and basic physical quantities. Operational quantities are used for direct measurement and include the ambient dose equivalent to that used in this work. They have been defined to provide a reasonable estimate of the second group, the radiation protection quantities. These are not directly measurable, but are used to estimate the health effects of ionizing radiation and are therefore used to set exposure limits. This includes the effective dose, for which limit values are set in current legislation. The relationship between these quantities is defined by relating these quantities to the basic physical quantities, fluence, kerma in the air and absorbed dose, by means of conversion factors. For photon radiation, the conversion factors are related to the kerma in the air. For each exposure situation assessed, the value of the measured operational quantity shall always be greater than, or at least equal to, the calculated value of the radiation protection quantity, while the disparity shall not be large. According to ICRU report No. 57 [20], for photon radiation in the energy range 60 keV–10 MeV the ratio between the effective dose and the ambient dose equivalent is in the range 0.75–0.92, and for low energy radiation this ratio is even lower. To demonstrate compliance with the limits expressed in radiation protection quantities, it is therefore possible to directly use the operational quantities as a conservative estimate.

While reducing the radiation dose per scan is particularly relevant in clinical and biological CT, the effects of X-ray exposure on the scanned object are often negligible in industrial CT [15]. General principles of radiation protection, justification, optimization and dose limitation [21] thus apply only to the potential public and occupational exposure.

The first principle of justification, to ensure that the individual or societal benefit resulting from the practice outweighs the health detriment that it may cause, can be documented by the benefit of optimization of log sawing based on the CT scan data. A large published review [22] states 3% to 28% lumber value improvement based on the knowledge of position and size of internal log defects, such as knots and decay, depending on the wood species. Such benefit can hardly be achieved using alternative methods while related radiation exposure risks can be optimized.

In the second principle of optimization, with the given source of ionizing radiation, several methods can be applied to optimize the exposure. The three main methods used to control the exposure to ionizing radiation are time, distance and shielding.

Typical CT scanning devices will have a shielding of the radiation source of some kind, to ensure the directional and energy properties of the emitting radiation. The scanning chamber itself will furthermore shield the radiation. Properties of the shielding can be optimized in order to achieve the desired shielding effect, while keeping in mind technical requirements such as access to the device.

Time is in many situations the most effective method to optimize the exposure. In case of industrial CT scanners, it would be applied as a combination of work planning and application of operating procedures for the workers involved. The scanning process is controlled remotely from the control room; the only relevant workplace is log manipulation.

Distance is applied directly with the use of a remote control and operation of the scanning unit. Furthermore, the concept of controlled area and supervised area allows to define areas where certain exposure can be expected and to apply relevant measures to control the presence of persons.

The third principle, dose limitation, is required by the relevant legislation. In the European Union according to Directive 2013/59/EURATOM, the limit on the effective dose for occupational exposure shall be 20 mSv in any single year [23]. This is often transposed to national legislation, as is the case for Slovakia [21].

This paper is focused on the issue of radiation protection of the wood log scanning line, which includes the unique computer tomography technology. This technology makes it possible to create a 3D model of a log of wood and display its internal defects. This model subsequently serves to optimize cutting plans in order to maximize yield. There are currently 14 CT log scanners installed in the world, and the 15th one mentioned in this study is operated in a research and development environment, where it serves as a tool for increasing the competitiveness of the forestry sector. The chosen form of the case study demonstrates a specific situation in operation and applies the general principles of radiation protection in practice. Currently, such a case study has not been published and can be of benefit to both the scientific and application communities.

The aim of this paper is to quantify the radiation exposure of workers at a 3D CT wood scanning workplace based on measurements of ionizing radiation levels during the operation of a log CT scanner, to compare it with limit values, and to determine a monitoring program.

## 2. Materials and Methods

### 2.1. Description of the 3D CT Scanning Workplace 

The workplace is located in the Biotechnology Park of the National Forest Centre. The ionizing radiation source is an X-ray tube placed in a protective cabin as a 3D CT device MICROTEC (Brixen, Italy). The CT scanner is part of the 3D CT scanning line (Figure 1) and its function is continuous quality scanning or detection of internal defects of the wood under examination. The examined logs are loaded mechanically on a conveyor with a feed speed of 5 m/min, which allows scanning the material along its entire length. The inlet and outlet openings of the CT scanner’s protective cabin are protected by slats of lead rubber declared by the manufacturer as the equivalent of 4 mm Pb at 225 kV.

After entering the protected scanning cabin through the scanning portal, the tree trunk is tomographically scanned. The results of the diagnostics are displayed directly on a PC in the control room. The vault is located in a specially protected area of the 3D CT scanning workstation.

The CT scanner is capable of measuring quality, deformation and other values through the penetration of X-rays with a voltage of up to 225 kV. A narrow fan-shaped moving beam of radiation is detected by the sensor array. The device can be accessed from both sides. Access is restricted and secured against unwanted entry by barriers with safety components. The operation is fully automatic, but the device requires the presence of an operator to monitor, control and intervene in the management of the operation system (Figure 1).

The workplace itself and the surrounding area are marked with a label, showing the symbol of the ionizing radiation source according to the relevant legislation of the country where the radiation source is operated.

### 2.2. Method

The measurement of leakage radiation during scanning was performed with a metrologically verified instrument. The measured value was the ambient dose equivalent rate ${\dot{H}}^{*}\left(10\right)$. The measurement was carried out on 7th of October 2022. In the first step, the ambient dose equivalent rate ${\dot{H}}^{*}\left(10\right)$ of the background was measured, i.e., with the CT scanner switched off. In the second step, measurement locations were selected at which the ${\dot{H}}^{*}\left(10\right)$ value was subsequently measured during the log scanning process. The following Figure 2 shows the layout of the 3D CT wood scanning workstation with the operator staff positions and measurement points M1 to M29 marked.

### 2.3. Methodology for Calculating the Effective Dose Value E

The effective dose E is the sum of the weighted equivalent H_T_ doses in all organs or tissues of the body due to internal and external exposure multiplied by the appropriate tissue weighting factor *W*_T_
(1)$$E=\underset{T}{\sum }W_{T}.H_{T}=\underset{T}{\sum }W_{T}.\underset{R}{\sum }W_{R}.D_{T.R}$$
where W_T_ is the tissue weighting factor of the tissue or organ T, W_R_ is the radiation weighting factor of the ionizing radiation R and *D*_T,R_ is the mean absorbed radiation dose R in the tissue T.

The tissue weighting factors W_T_ of an organ or tissue T represent the relative contribution of that organ to the total health damage caused by the stochastic effects of ionizing radiation. The unit of the effective dose is Sievert (Sv). One Sievert is equivalent to one joule per kilogram.

### 2.4. Measuring Equipment

Two independent measuring devices were used for the measurement of the leakage radiation. Both devices are calibrated. Full specifications of the measuring devices are listed in the Table 1.

APVL Thermo Scientific (Saint-Cyr-sur-Loire, France), FH 40 G-L10 Ω ambient dose equivalent rate meter ${\dot{H}}^{*}\left(10\right)$: Measurement working range for of 10 nSv/h–100 mSv. Energy range 30 keV–4.4 MeV. Data error: typically <5%, maximum 20%, for ^137^Cs radiation (E = 662 keV). Directional dependence <20% over the range −75°, +75° with regard to the longitudinal axis of the instrument.

Thermo Scientific, RadEye G 20-10 (Loughborough, UK), ambient dose equivalent rate meter ${\dot{H}}^{*}\left(10\right)$: Measuring working range for dose equivalent 10 nSv/h–2 mSv. Energy range 17 keV–1.3 MeV. Data error: typically <5%, maximum 20%, for ^137^Cs radiation (E = 662 keV). The instrument is equipped with a telescopic rod, for making measurements in inaccessible places.

Measurement quality was ensured in several ways. High-end measuring devices designed for this purpose were used for the measurement. The measuring devices were calibrated by competent authorities and metrologically verified. The measurement was carried out by a person with the appropriate accreditation. The measurement itself took place simultaneously on two measuring devices, with one measuring device serving as a control for the measured values.

### 2.5. The Source of Radiation

The source of ionizing radiation was an MXR-225FB X-ray lamp placed in the rotating portal for the purpose of continuous CT scanning of tree trunks. CT scans of tree trunks were taken in order to detect internal wood defects. The following Table 2 lists the technical parameters of the radiation source.

Figure 3 shows the radiation source, an X-ray tube placed in a rotating portal that rotates around the object being scanned.

## 3. Results

The measured background value of the ambient dose equivalent rate at the 3D CT wood scanning workstation is 0.13 μSv/h. Table 3 shows the measured values of the ambient dose equivalent rate ${\dot{H}}^{*}\left(10\right)$ at different workplace locations of the 3D CT wood scanning workstation compared with the occupational exposure limit. Table 4 shows the measured values of the ambient dose equivalent rate ${\dot{H}}^{*}\left(10\right)$ at all measured locations of the 3D CT wood scanning workstation, compared to the exposure limit value for non-exposed outside workers as well as for the supervised area. The CT equipment was in working mode during the measurement—scanning of logs with a diameter of 35 cm was in progress.

The measured values show that the ambient dose equivalent rate at measurement points M1, M28 and M29 was the same as the ambient dose equivalent rate of the background. Measurement sites M26 and M27 showed an increased value of 0.02 μSv/h and 0.04 μSv/h compared to the background.

The highest ambient dose equivalent rate values were obtained at measurement points M4 and M5, which are the log inlet and outlet of the CT scanner, and measurement points M14 and M20, which are the interfaces between the shielded part of the CT scanner container and the inlet and outlet scanning tunnels. As the distance from these points increases, the measured ambient dose equivalent rate values decrease (measurements from M14 to M19 and measurements from M20 to M25).

Based on the measured values of the ambient dose equivalent ${\dot{H}}^{*}\left(10\right)$ at the measurement points, a 3D CT radiation exposure map of the log scanning workplace was interpolated (Figure 4).

Given the ambient dose equivalent rate values reached at the measurement sites that are accessible to non-exposed outside workers during CT scanner operation, there is a risk that they could be exposed to ionizing radiation that comes close to or exceeds the limit (see Table 5 for a comparison with the annual limit of 1 mSv effective dose). It is therefore necessary to establish safety measures to prevent people accessing these sites. These were implemented in the form of a fence defining the supervised area (Figure 5). The legislation requirements for establishment of a supervised area in the Slovak legislation [21] are defined as a possibility to exceed the annual effective dose 1 mSv. Subsequently, control measurements of the ambient dose equivalent rate at the boundary of the fence were made at measurement points M30 to M35.

Table 5 shows the measured ambient dose equivalent rate values at the fence boundary.

The results show that the implemented safety measures (establishment of supervised area) fulfil their function, since no person can be exposed to ionizing radiation exceeding the annual limit of 1 mSv effective dose per year at any point accessible to outside workers during the operation of the CT scanner. To the above results it should be noted that the doses were calculated for a 500 h time exposure for the outside workers. This exposure time is conservatively overestimated. The overestimation is based on the fact that the CT scanner will not be operating continuously and the expected working time of the X-ray tube will not exceed the 500 h estimate. Furthermore, it is neither expected, nor necessary, for any person to stay in the area near the protective fence during the operation of the CT scanner. In reality, the real exposure time will be considerably shorter and thus the results of the received dose at the individual measurement points will be lower. The real annual exposure time will be monitored in the future and the results will be re-evaluated according to the observed data.

## 4. Discussion

The operating CT scanner for logs of wood is comparable in design to CT scanners used in medicine. The difference is in the adaptation of the dimensions of the device so that the wood logs can be scanned. In terms of physical principle, the operation of the CT scanner is the same as that of medical CT scanners. In contrast to the use of CT scanners in medicine, when scanning logs of wood we do not have to take the need to minimize the exposure of the scanned subject to ionizing radiation into account. A CT log scanner can operate at full power, which is approximately double that of a medical CT scanner. A medical CT scanner has a limit for X-ray voltages of 120 to 130 kV, in the case of an industrial CT scanner the X-ray voltages can reach 225 kV [7].

The International Commission on Radiological Protection (ICRP) provides guidance and recommendations on radiation protection. When it comes to industrial CT scanner use, several ICRP publications may be relevant. The ICRP Publication 103 (2007)—“The 2007 Recommendations of the International Commission on Radiological Protection.” [24]—provides the foundational recommendations on radiation protection, including dose limits and principles for radiation protection. It lays the groundwork for understanding radiation protection in various contexts, including industrial applications. The ICRP Publication 129 (2021)—“Radiological Protection in Cone Beam Computed Tomography.” [25]—may be particularly relevant to industrial CT scanner use, as it addresses the radiation protection aspects of cone beam computed tomography, a technology similar to traditional CT scanning. The ICRP Publication 75,—“General Principles for the Radiation Protection of Workers.” [26]—provides fundamental principles for radiation protection and is applicable to various contexts, including industrial uses of radiation-emitting devices like CT scanners. While it may not be specific to CT scanners, it offers general guidelines for the protection of workers in radiation environments.

In the protection of individuals, whether residents or workers, the protection against ionizing radiation in the operation of an industrial CT scanner is very effective. A full range of protective features such as spacing, demarcation of space and shielding barriers can be used to effectively prevent the penetration of ionizing radiation. As a result, the industrial CT scanning worker may not be exposed to ionizing radiation above the background level.

There are risks associated with a failure or crash of the CT scanner. The risk of radiation exposure in the case of CT scanners is minimized by the very nature of the radiation source. The X-ray tube is a source of radiation only if it is powered by electricity. In the case of a failure or breakage of the safety barrier, the electric current is interrupted and the X-ray tube becomes harmless from the aspect of the risk of radiation exposure. This same advantage is also present if the X-ray tube is stolen. It can be concluded that the operation of a CT scanner is less risky than the operation with a radionuclide radiation source in terms of control of the radiation source.

Today, there are many professional and scientific articles available that focus on the issue of radiation risk from computed tomography. These publications are mainly focused on the area of medical use and risk optimization for patients undergoing such tests. The modification of CT scanner performance for each patient was addressed by Bora et al., 2014 [27]. Optimization of radiation exposure from CT scans performed in pediatrics has been addressed by Brody et al., 2007, and Frush et al., 2003 [28,29].

In order to ensure a high level of radiation protection at the CT scanner workstation, the type and energy spectrum of the radiation must be taken into account. In the area to be monitored this is scattered X-ray radiation. The energy spectrum depends on the shielding material, the voltage on the electrode and the low-energy radiation filters used [22].

There are few publications focusing on the safety of CT scanning in industry. Zhou et al., 2016, conducted a study on the radiation protection of industrial computed tomography workstations [30]. In a large book publication, Industrial X-ray Computed Tomography, by Carmignato et al., 2018, the issue of safety and risk associated with CT scanner operation is not covered [31]. For these reasons, this case study can provide valuable information for the design and provision of other industrial computed tomography workplaces from the perspective of human protection from ionizing radiation.

## 5. Conclusions

For the purpose of assessment and subsequent optimization of the effective dose of ionizing radiation at the workplace of 3D CT scanning of logs, the individuals assessed were a professional representative from the number of workers working with the ionizing radiation source and a maintenance person from the number of workers working with ionizing radiation sources. On the basis of the results, it can be concluded that for an operation of 500 h per year with an ionizing radiation source with an ambient dose equivalent rate of approximately 0.19 μSv/h, the limit value of 1 mSv annual effective dose for a non-exposed classified worker working with ionizing radiation sources will not be exceeded in normal operation. The above applies only in the case of staying and working at the workplace workstations and locations outside the CT scanner log fence.

For the purpose of assessment and subsequent optimization of the effective dose of ionizing radiation at the 3D CT scanning workplace for the population, we consider an ambient dose equivalent rate value of about 0.02 μSv/h (subtracting terrestrial and cosmic background) at a distance of 3 m from the CT scanner for 500 h per year. The resulting effective dose to the resident is less than 10 μSv/year.

The above assessment of the radiation exposure of the 3D CT scanning of logs to workers and the general public during operation shows that it will not be necessary to demonstrate optimization of radiation protection for this activity. When comparing the cost of additional protection from ionizing radiation with the benefit of these measures, the cost of an adequate protective barrier (Pb, Fe) that would have to cover the CT scanner logs significantly outweighs the benefits of the additional protective barrier (Pb, Fe). The measurement has proven that the existing shielding is sufficient to achieve a high level of radiation protection. In addition to increasing costs, additional shielding would also limit the technological performance in terms of accessibility of the device for inspection and maintenance, thus reducing the overall efficiency of the operation of the 3D CT scanner for logs.

## Figures and Tables

**Figure 1 sensors-23-08937-f001:**
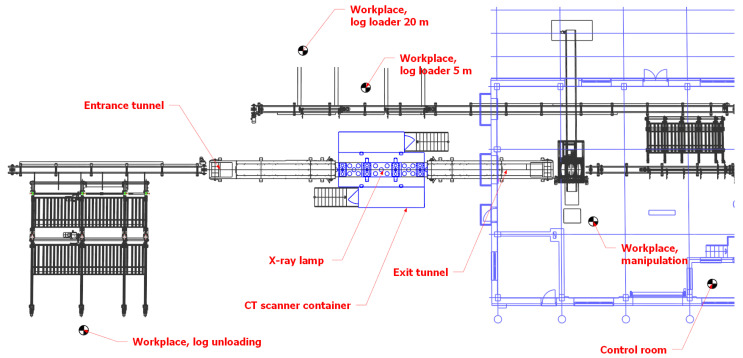
Schematic of the 3D CT scanning line.

**Figure 2 sensors-23-08937-f002:**
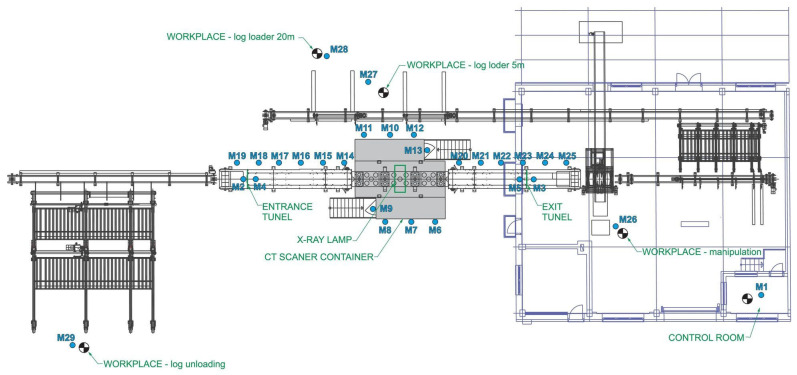
Layout of the 3D CT wood scanning workplace and selected measuring points M1 to M29.

**Figure 3 sensors-23-08937-f003:**
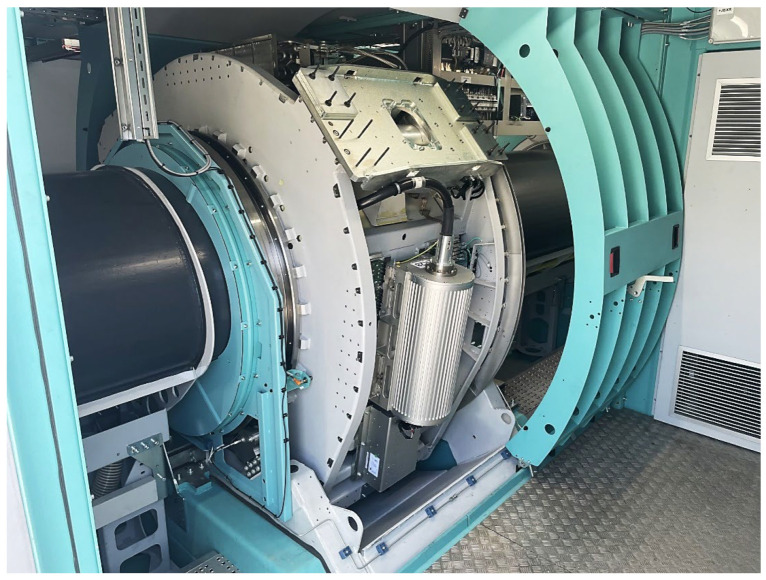
Radiation source located in the rotating portal of a CT scanner.

**Figure 4 sensors-23-08937-f004:**
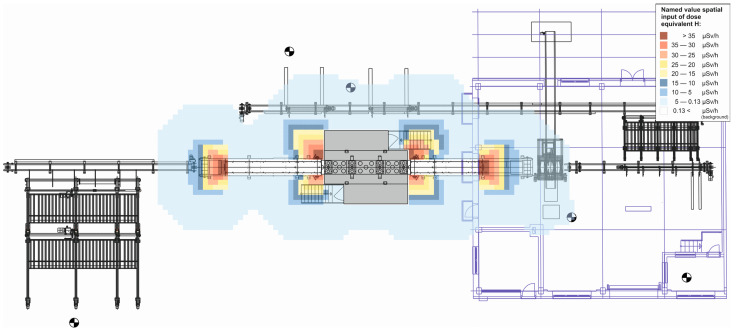
Map of ionizing radiation exposure of a 3D CT scanning workstation for logs.

**Figure 5 sensors-23-08937-f005:**
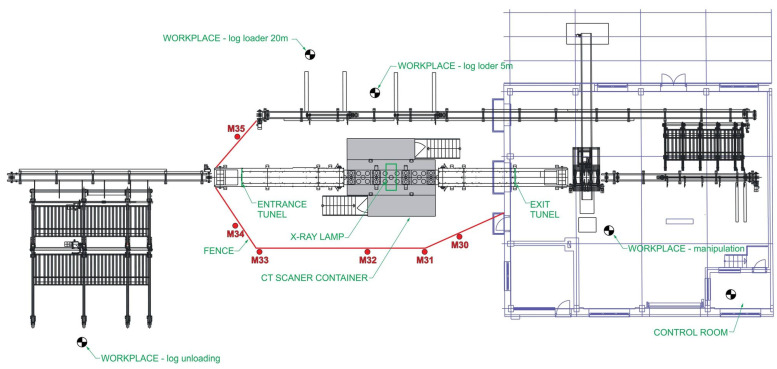
Definition of the supervised area by fencing—red.

**Table 1 sensors-23-08937-t001:** Technical parameters of the measuring devices.

Device	APVL Thermo Scientific, FH 40 G-L10 Ω	Thermo Scientific, RadEye G 20-10
Type of device	Multi-purpose digital survey meter	X-ray and gamma survey meter
Type of detector	Proportional counter	G-M counter with advanced digital filter (ADF)
Measured quantity	Sievert (Sv; ambient dose equivalent rate ${\dot{H}}^{*}\left(10\right)$)	
Measurement working range	10 nSv/h–100 mSv	10 nSv/h–2 mSv
Energy range	30 keV–4.4 MeV	17 keV–1.3 MeV
Data error	Typically < 5%, maximum 20%, for ^137^Cs radiation (E = 662 keV)	Typically < 5%, maximum 20%, for ^137^Cs radiation (E = 662 keV)

**Table 2 sensors-23-08937-t002:** Technical parameters of the radiation source.

Facility	MICROTEC CT.LOG X-ray Computer Tomography Scanner
X-ray lamp	MXR-225FB
X-ray lamp cover	MOR—225FBC
Rated voltage	225 kV
Electric current	13 mA
Incandescent current	4.2 A
Power	3 kW
Focal point according to EN 1254	d = 5.5 mm, material is tungsten
Angle of the target	20 degrees
Cooling medium	water
Scanning speed	5 m/min

**Table 3 sensors-23-08937-t003:** Measured ambient dose equivalent rates at workstation sites compared with occupational exposure limit.

Designation of the Measuring Point	Description of the Measuring Point	$\mathrm{Measured}\;\mathrm{Value}\;\mathrm{Ambient}\;\mathrm{Dose}\;\mathrm{Equivalent}\;\mathrm{Rate}\;{\dot{H}}^{*}\left(10\right)$	Comparisons with 20 mSv/year Effective Dose *E* Limit for an Exposure Duration of 2000 h/year
M1	Control room	0.13 μSv/h	0.260 mSv 1.3% of the limit
M26	Handler’s workstation, inside the building	0.17 μSv/h	0.340 mSv 1.7% of the limit
M27	Handler’s workstation, log loading 5 m	0.15 μSv/h	0.300 mSv 1.5% of the limit
M28	Handler’s workstation, loading logs with a loader 20 m	0.13 μSv/h	0.260 mSv 1.3% of the limit
M29	Handler’s workstation, log unloading 7 m	0.13 μSv/h	0.260 mSv 1.3% of the limit

**Table 4 sensors-23-08937-t004:** Measured ambient dose equivalent rates at selected measurement sites compared with supervised area effective dose limit.

Designation of the Measuring Point	Description of the Measuring Point	$\mathrm{Measured}\;\mathrm{Value}\;\mathrm{Ambient}\;\mathrm{Dose}\;\mathrm{Equivalent}\;\mathrm{Rate}\;{\dot{H}}^{*}\left(10\right)$	Comparisons with 1 mSv/year Effective Dose *E* Limit for an Exposure Duration of 500 h/year
M1	Control room	0.13 μSv/h	0.063 mSv 6.3% of the limit
M2	Input to CT scanner, surface of slats	1.75 μSv/h	0.848 mSv 84% of the limit
M3	Output to CT scanner, surface of slats	1.30 μSv/h	0.631 mSv 63% of the limit
M4	In the CT scanner tunnel—entrance	65 μSv/h	31.525 mSv 32 times the limit
M5	In CT scanner tunnel—exit	33 μSv/h	16.005 mSv 16 times the limit
M6	Surface of the cabin at the X-ray source location	0.20 μSv/h	0.097 mSv 9.7% of the limit
M7	Cabin surface section left	0.14 μSv/h	0.068 mSv 6.8% of the limit
M8	Cabin surface section right	0.14 μSv/h	0.097 mSv 9.7% of the limit
M9	Cabin door surface	0.23 μSv/h	0.112 mSv 11.2% of the limit
M10	Cabin surface at X-ray source location	0.25 μSv/h	0.121 mSv 12.1% of the limit
M11	Cabin surface section left	0.20 μSv/h	0.097 mSv 9.7% of the limit
M12	Cabin surface section right	0.20 μSv/h	0.097 mSv 9.7% of the limit
M13	Cabin door surface	0.22 μSv/h	0.107 mSv 10.7% of the limit
M14	First section entrance tunnel	30 μSv/h	14.55 mSv 15 times the limit
M15	Second section entrance tunnel	14 μSv/h	6.79 mSv 7 times the limit
M16	Third section entrance tunnel	5.6 μSv/h	2.716 mSv 2.7 times the limit
M17	Fourth section entrance tunnel	4.1μSv/h	1.989 mSv 2 times the limit
M18	Fifth section entrance tunnel	2.3 μSv/h	1.116 mSv 1.12 times the limit
M19	Sixth section entrance tunnel	1.2 μSv/h	0.582 mSv 58.2% of the limit
M20	First section exit tunnel	40 μSv/h	19.4 mSv 19.4 times the limit
M21	Second section exit tunnel	17 μSv/h	8.25 mSv 8.25 times the limit
M22	Third section exit tunnel	6.2 μSv/h	3.00 mSv 3 times the limit
M23	Fourth section exit tunnel	4.1 μSv/h	1.99 mSv 1.99 times the limit
M24	Fifth section exit tunnel	2.1 μSv/h	1.02 mSv 1.02 times the limit
M25	Sixth section exit tunnel	1.1 μSv/h	0.534 mSv 53.4% of the limit
M26	Handler’s workstation, inside the building	0.17 μSv/h	0.085 mSv 8.5% of the limit
M27	Handler’s workstation, log loading 5 m	0.15 μSv/h	0.073 mSv 7.3% of the limit
M28	Handler’s workstation, loading logs with a loader 20 m	0.13 μSv/h	0.063 mSv 6.3% of the limit
M29	Handler’s workstation, log unloading 7 m	0.13 μSv/h	0.063 mSv 6.3% of the limit

**Table 5 sensors-23-08937-t005:** Measured ambient dose equivalent rate values at the CT scanner fence boundary.

Designation of the Measuring Point	Description of the Measuring Point	$\mathrm{Measured}\;\mathrm{Value}\;\mathrm{Ambient}\;\mathrm{Dose}\;\mathrm{Equivalent}\;\mathrm{Rate}\;{\dot{H}}^{*}\left(10\right)$	Comparisons with 1 mSv/year Effective Dose *E* Limit for an Exposure Duration of 500 h/year
M30	Protective fence, first section	0.45 μSv/h	0.218 mSv 21.8% of the limit
M31	Protective fence, second section	0.43 μSv/h	0.208 mSv 20.8% of the limit
M32	Protective fence, third section	0.43 μSv/h	0.208 mSv 20.8% of the limit
M33	Protective fence, fourth section	0.41 μSv/h	0.199 mSv 19.9% of the limit
M34	Protective fence, fifth section	0.40 μSv/h	0.194 mSv 19.4% of the limit
M35	Protective fence, sixth section	0.40 μSv/h	0.194 mSv 19.4% of the limit

## Data Availability

Not applicable.

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
