# Peer review of "Radiation Protection of a 3D Computer Tomography Scanning Workplace for Logs—A Case Study"

_sensors, 2023, doi:10.3390/s23218937_

Round 1

Reviewer 1 Report

Comments and Suggestions for Authors

Although the manuscript is clearly and well written it is more like a technical report not a scientific paper. I would expect at least description of possible solution how to minimaise the radiation (and not in the conclusion section only).

The Directive related to dose limits should be mentioned – the limit of 1 mSv is typical for the population, for the workers it is usually higher! If the local law is not different it means that discussion is incorrect.

Line 71-73 – it is not clear if Authors mean personnel or patients. Patients sometimes receive relatively high doses, but this is still a benefit, to diagnose the disease.

Quality of measurements is not discussed.

Comments on the Quality of English Language

no comments

Author Response

Thank you very much for the comments of Reviewer 1 and our responses are attached.

Reviewer 2 Report

Comments and Suggestions for Authors

All comments are reported in the attached file.

Please, pay attention and consider all the main comments and suggestions.

Comments on the Quality of English Language

Minor editing of English language required. Make uniform the words used in the text.

Author Response

Thank you very much for the comments of Reviewer 2 and our responses are attached.

Round 2

Reviewer 1 Report

Comments and Suggestions for Authors

The manuscript improved and I am satisfied with corrections. However, I still find it like a technical report not a scientific paper. Therefore, the article should be preceded by an appropriate annotation.

Author Response

Thank you forReviewer 1 comments and our responses are attached
